# *L*-Lactate Administration Improved Synaptic Plasticity and Cognition in Early 3xTg-AD Mice

**DOI:** 10.3390/ijms26041486

**Published:** 2025-02-11

**Authors:** Yaxin Wang, Jinfeng Zhao, Li Zhao

**Affiliations:** 1School of Physical Education, Shanxi University, Taiyuan 030006, China; wyx199585@163.com (Y.W.); zjf@sxu.edu.cn (J.Z.); 2Key Laboratory of Physical Fitness and Exercise, Ministry of Education, Beijing Sport University, Beijing 100084, China

**Keywords:** *L*-lactate, AD, synaptic plasticity, cognition

## Abstract

Synaptic plasticity impairment and behavioral deficits constitute classical pathological hallmarks in early-stage Alzheimer’s disease (AD). Emerging evidence suggests these synaptic dysfunctions may stem from metabolic dysregulation, particularly impaired aerobic glycolysis. As a key product of astrocyte-mediated aerobic glycolysis, lactate serves dual roles as both an energy substrate and a signaling molecule, playing a critical regulatory role in synaptic plasticity and long-term memory formation. This study investigated whether exogenous *L*-lactate supplementation could ameliorate synaptic dysfunction and cognitive deficits in early-stage AD models. Our findings reveal significant reductions in hippocampal lactate levels in experimental AD mice. Systemic administration of *L*-lactate (200 mg/kg) effectively restored physiological lactate concentrations in both hippocampal tissue and cerebrospinal fluid (CSF). Chronic *L*-lactate treatment significantly improved spatial learning and memory performance in behavioral assessments. Electrophysiological recordings demonstrated that either acute bath application of *L*-lactate (2 mM) to hippocampal slices or chronic intraperitoneal administration enhanced high-frequency stimulation (HFS)-induced long-term potentiation (LTP) magnitude in 3xTg-AD mice. Ultrastructural analysis revealed that *L*-lactate treatment enhanced synaptic density and improved morphological features of hippocampal synapses. At the molecular level, *L*-lactate administration upregulated synaptic marker synaptophysin (SYP) expression while downregulating activity-regulated cytoskeletal-associated protein (ARC) levels in AD mice. These multimodal findings demonstrate that exogenous *L*-lactate supplementation effectively restores synaptic plasticity and cognitive function in early-stage 3xTg-AD mice through concurrent improvements at behavioral, structural, and molecular levels.

## 1. Introduction

Alzheimer’s disease (AD) is a neurodegenerative disease characterized by progressive memory decline and cognitive impairment. The loss or damage of synapses is central to the disease process, which leads to neuronal dysfunction and subsequent neuronal apoptosis [1]. Synaptic activity requires a sufficient amount of energy to maintain; a lack of energy initiates and worsens the synaptic pathology in AD over time.

Aerobic glycolysis (AG), a crucial metabolic pathway in the brain, not only generates energy but also supports biosynthesis and neuroprotection. Emerging evidence has demonstrated significant alterations in brain AG during early AD pathogenesis, with reduced glycolytic flux correlating strongly with dementia severity. Notably, the spatial distribution of amyloid-β (Aβ) plaques and hyperphosphorylated tau deposits shows remarkable correspondence with regional AG impairment patterns [2,3]. Furthermore, the metabolic significance of AG extends to neurodevelopmental processes, as evidenced by the colocalization of high AG activity regions with gene expression profiles governing synaptic plasticity, neurite outgrowth, and neural circuit remodeling [4]. These collective findings position AG modulation as a promising therapeutic strategy for enhancing synaptic resilience and potentially halting AD progression.

One of the mechanisms by which AG improves synaptic plasticity in AD is through the entry of its products into neurons as neurotransmitters or energy substrates [5]. Lactate, synthesized from astrocytic aerobic glycolysis, has long been considered as providing an energy substrate to fuel neurons’ high energy demands [6,7] and being a signaling molecule for synaptic plasticity [8,9]. Neurophysiological studies have revealed sensory and cognitive stimuli in humans and animals produce an increase in brain lactate [10,11], with exogenous lactate administration demonstrating therapeutic potential in reversing memory deficits and LTP impairments associated with glycogen metabolism dysfunction. In the early stage of AD, lactate concentration was decreased in the cerebral cortex and hippocampus of AD mice [12]. In addition, lactate production via the aerobic glycolytic pathway is altered in 3xTg AD astrocytes [13].

Accumulating evidence suggests that lactate supplementation may ameliorate cognitive deficits induced by brain injury [14] and exert antidepressant effects [15]. However, in the context of AD, acute exogenous lactate administration has demonstrated limited efficacy in improving global cognitive function among AD patients, with only marginal enhancements observed in semantic memory [16,17]. Given that the early stages of AD represent a critical therapeutic window for intervention, it remains unclear whether chronic lactate supplementation during this pivotal phase could mitigate cognitive impairments associated with aerobic glycolysis dysfunction. Thus, in this study, we investigated whether lactate could improve cognition, synaptic structure, and function in early 3xTg-AD mice and tested the hypothesis that lactate may be a potential target for the prevention and treatment of AD by improving synaptic plasticity and memory.

## 2. Results

### 2.1. Exogenous L-Lactate Increased the Concentration of Brain Lactate in 3xTg-AD Mice

Lactate plays a key role in synaptic plasticity and long-term memory formation, and the hippocampus is a key structure associated with the pathophysiology of memory loss in AD; therefore, we examined lactate concentrations in CSF and hippocampal tissues of 6-month-old 3xTg AD mice. Compared to the age-matched wild-type C57BL/6J controls (CS group), the 3xTg-AD mice exhibited significantly reduced lactate levels in hippocampal tissues (*p* = 0.038), though CSF lactate concentrations remained unchanged. To evaluate lactate’s therapeutic potential, we administered intraperitoneal *L*-lactate (200 mg/kg; AL group) or vehicle (NaCl; AN group) to AD mice. Within 30 min post injection, *L*-lactate treatment elevated CSF lactate by approximately 0.6 mmol/L (*p* = 0.014) and significantly restored hippocampal lactate concentrations (*p* = 0.047) (Figure 1).

### 2.2. Effect of L-Lactate on Cognition in 3xTg-AD Mice

Studies have identified that lactate was required for long-term memory formation; thus, can memory dysfunction in AD mice be delayed or prevented from occurring after *L*-lactate supplementation? AD mice received daily injections of *L*-lactate or NaCl for 4 weeks before initiating a battery of behavioral tests (Figure 2A). Spatial learning performance, evaluated via an eight-arm radial maze, revealed that vehicle-treated AD mice committed significantly more reference memory errors than age-matched wild-type controls (*p* = 0.001). While chronic treatment with *L*-lactate significantly reduced the frequency of reference memory errors (*p* = 0.009) and increased the percentage of correct choices (*p* = 0.031) in AD mice (Figure 2C–G), indicating improved hippocampal-dependent memory. Fear-associated memory was assessed through a step-down passive-avoidance test; no difference was found between the three groups in the latency time 48h after training (Figure 2F,G).

### 2.3. Effect of L-Lactate on LTP Magnitude in 3xTg-AD Mice

Early studies have demonstrated that HFS protocols are used to induce LTP to mimic the synchronized firing of neuronal ensembles observed during natural learning processes and are directly involved in memory encoding. As such, LTP has been widely adopted as a canonical cellular model of memory formation, serving as a quintessential experimental paradigm for investigating synaptic plasticity [18]. Firstly, we added *L*-lactate (2 mM) to aCSF, and, after 10 min of incubation, LTP was induced by HFS; we found that L-lactate treatment significantly enhanced LTP magnitude (*p* = 0.000) (Figure 3A,E). Following that, we compared the effects of 2 mM and 5 mM lactate concentrations on LTP magnitude. Intriguingly, increasing the lactate concentration to 5 mM did not further enhance LTP compared to 2 mM (Figure 3B,E). In addition, we blocked lactate trafficking to neurons by inhibiting monocarboxylate transporters using AR-C155858 (blocking monocarboxylate transporters (MCT1/2). We found that the enhancement of L-lactate on LTP was abrogated by AR-C155858 (*p* = 0.050) (Figure 3C,E). Next, we examined whether pyruvate, a downstream glycolytic metabolite of lactate, could improve LTP similarly to lactate. The results indicated pyruvate (2 mM) did not improve LTP magnitude in brain slices of AD mice as effectively as lactate (*p* = 0.000) (Figure 3D,E). Mirroring these in vitro findings, chronic in vivo L-lactate supplementation (4 weeks) significantly improved LTP magnitude in 6-month-old 3xTg-AD mice compared to AD controls (*p* = 0.013) (Figure 3F,G).

### 2.4. Effect of L-Lactate on Synaptic Structure in 3xTg-AD Mice

Defects in synaptic structure and function are major events in the early stages of AD that lead to impaired synaptic transmission; therefore, we assessed the ultrastructural changes of the synaptic structure of each group of mice by transmission electron microscopy (Figure 4A). Compared with CS mice, the number of synapses was dramatically decreased in AD mice (*p* = 0.028). However, exogenous *L*-lactate supplementation significantly increased the total number of synapses (*p* = 0.002) (Figure 4B). In addition, the thickness of postsynaptic density (*p* = 0.001) and the width of the synaptic cleft (*p* = 0.000) were significantly reduced in the AN group; after 4 weeks of *L*-lactate supplementation, both were significantly improved (*p* = 0.000) (Figure 4C,D). It is worth noting that there was no observable difference in synaptic curvature (Figure 4E). Taken together, these results indicated that the exogenous *L*-lactate supplementation could effectively improve synaptic structure in AD mice.

### 2.5. Effect of L-Lactate on the Expression of Synaptic Plasticity-Related Proteins in 3xTg-AD Mice

In addition, we tested whether *L*-lactate influenced neuronal hippocampal molecular changes known to be associated with synaptic plasticity in AD mice. To this end, we detected the expression of major intermediates of plasticity-related processes, such as postsynaptic density protein 95 (PSD95), synaptophysin (SYP), and immediate early genes, such as early growth response 1 (Egr1) and activity-regulated cytoskeletal-associated protein (Arc) (Figure 5A). Quantitative Western blot analyses indicated that, compared with the CS group, the expression of SYP (*p* = 0.034) and EGR1 (*p* = 0.03) was significantly decreased in the AN group. In contrast, exogenous *L*-lactate increased the expression of SYP (*p* = 0.005) and decreased the expression of ARC (*p* = 0.001) compared to the AN group (Figure 5B–E).

## 3. Discussion

Our study found that the concentration of lactate in the hippocampus decreased in early AD mice. Acute application of *L*-lactate in AD brain slices improved LTP. Furthermore, exogenous supplementation of *L*-lactate alleviated deficits in learning and memory, enhanced LTP magnitude, and improved synaptic ultrastructure and synaptic plasticity-related protein expression in AD mice.

Impairment of brain energy metabolism in AD leads to changes in lactate concentration, but the results of lactate concentration in AD hippocampal tissue and CSF have yielded inconsistent results. Animal experiments have shown that the lactate levels in the cerebral cortex and hippocampus of Aβ_25-35_-treated rats were decreased compared with those in the control rats [19], and similar results were observed in 3-month-old APP/PS1 transgenic mice [12]. In population studies, a study by Bonomi et al. demonstrated that major neurodegenerative dementias like AD and FTD showed physiological but decreased levels of CSF lactate and bound the reduction of CSF lactates to the progression of tau pathology [20]. Another study showed that patients with mild cognitive decline showed higher CSF lactate levels than moderate-severe AD, and CSF lactate concentrations seemed to decrease in parallel with cognitive dysfunction, showing a weak positive correlation with MMSE scores [21]. Conversely, others suggested that CSF lactate concentrations were significantly elevated in patients with AD [22,23], and this increase may be a form of metabolic compensation triggered during neurodegeneration [24]. CSF lactate is highly dynamic and relies on concentration gradients and monocarboxylate transporters to enter neurons. In addition, age-dependent changes in metabolism and the integrity of the blood-brain barrier (BBB) may also contribute to the inconsistency of such results [25,26]. In addition, the inverted U-shaped trajectory of lactate dynamics in AD is mechanistically linked to stage-specific pathological alterations. During early AD pathogenesis, impaired cerebral energy metabolism—particularly the disruption of astrocytic aerobic glycolysis—leads to diminished lactate production [27], thereby exacerbating neuronal bioenergetic deficits. As AD progresses, hyperactivated microglia drive a pathological cascade, inducing reactive astrocyte transformation with amplified glycolytic flux (astrocytic metabolic reprogramming), which culminates in aberrant lactate overproduction [28]. This biphasic regulatory pattern may reflect stage-specific interactions between glial metabolic reprogramming and neurodegenerative progression, offering mechanistic insights into the dual roles of lactate in AD pathophysiology.

Recent studies have clearly shown the potential benefits of therapeutic lactate infusion under certain pathological circumstances, including rather common clinical conditions like traumatic brain injury. Lactate infusion (i.v.) was started 30 min after lateral fluid percussion injury and continued for 3 h. The results indicated that the lactate-infused injured animals demonstrated a significant improvement in cognitive ability compared to saline-infused injured animals [29]. The peripheral administration of L-lactate could produce antidepressant-like effects in different animal models of depression by modulating the expression of proteins involved in serotonin receptor trafficking, astrocyte functions, neurogenesis, nitric oxide synthesis, and cAMP signaling [15]. In addition, a study of AD patients given a single intravenous injection of sodium lactate solution for 20 min found that sodium lactate failed to significantly improve semantic categorization processes but slightly improved semantic memory [16]. Járdánházy et al. found that no significant changes were seen in the seriously damaged areas; the moderately affected regions in the close neighborhood showed a paradoxical inactivation with electroencephalographic slowing, and in areas not affected by neurodegenerative processes, synchronization likelihood decreased in theta bands and increased in the beta bands, which could be interpreted as a sign of improved functionality after lactate. However, these studies examined the effect of a single intravenous injection of sodium lactate solution for 20 min in patients with moderate dementia syndrome or late-onset sporadic AD on cognitive function or EEG power; it is not clear whether longer or multiple supplementations improve cognitive function in early AD [17]. In our study, we administered *L*-lactate supplementation to early AD model mice for 4 weeks and found that exogenous *L*-lactate supplementation reduced the frequency of reference memory errors, indicating improvements in the formation of long-term memories. However, it had no significant impact on the frequency of working memory errors (short-term memory) or avoidance latency (fear memory).

Hippocampal LTP maintenance requires glycogenolysis and is mediated by lactate. MCT2 will be saturated at lactate levels above ~2 mmol/L; therefore, we added 2 mM *L*-lactate to aCSF before HFS, and we found that *L*-lactate enhanced the maintenance of LTP in hippocampal slices of AD mice. Then, we further explored whether the *L*-lactate concentration effect could influence LTP magnitude; the results showed that 5 mM *L*-lactate did not have a more significant difference than 2 mM. However, it has been shown that elevated lactate (10 mM) reversibly suppressed the firing of hippocampal pyramidal cells [30], and resulted in the suppression of gamma oscillations and the attenuation of synaptic transmission and the reduction of neurotransmitter release [31]. AR-C155858, a monocarboxylate receptor inhibitor, blocked the translocation of lactate from astrocytes to neurons. Our results showed that AR-C155858 inhibited the ameliorative effect of *L*-lactate on the LTP of AD, which highlights that lactate plays a key role in synaptic plasticity in AD. It has been shown that glucose also has an effect in rescuing 1,4-dideoxy-1,4-imino-D-arabinitol (DAB)-induced memory impairment, but it was transient and only immediately after the injection, but not later [32]. This result may suggest that the primary mechanism of action of *L*-lactate on plasticity mechanisms was independent of its ability to act as an energy substrate. In addition, several specifically designed experiments consistently demonstrated that the excitatory effect of *L*-lactate was not due to its uptake by LC neurons and use as an energy substrate but acts on a substrate located on the extracellular side of their plasma membrane to excite them. Thus, we wanted to explore which role lactate plays in improving synaptic plasticity in AD mice. Pyruvate is a downstream metabolite of lactate and can be taken up by MCTs as one of the important energy substances for neurons. We found no improvement in LTP magnitude in AD mice after the addition of pyruvate (2 mM) before HFS. However, research has indicated that, during glucose deprivation, pyruvate and lactate not only maintained basal synaptic transmission but also allowed LTP induction [7]. Giannina et al. showed that pyruvate and the ketone body B3HB can functionally replace lactate in rescuing memory impairment caused by the inhibition of glycogenolysis or the expression knockdown of glia MCT1 and four in the dorsal hippocampus of rats [33]. We explored the reasons for the inconsistent results and found that these studies used a higher concentration of pyruvate (slices treated with 10 or 20 mM pyruvate in Yukitoshi’s study and an injection of 100 mmol/L in Giannina’s study), whereas we used a concentration of 2 mM. Supra-physiological concentrations of injected pyruvate may have been necessary to involve a sufficient tissue volume and duration to attain measurable behavioral effects, but this does not validate their relevance to normal physiology, and it has deleterious effects on neurons. Moreover, previous studies have utilized healthy rats to simulate memory loss through glucose depletion or glycogenolysis inhibition. However, in AD mice, memory deficits may arise from multiple factors, including impaired cerebral energy metabolism. The inability of 2 mM pyruvate to rescue LTP in AD mice suggests that lactate’s beneficial effects on synaptic plasticity extend beyond its role as an energy substrate, potentially involving unique signaling mechanisms. The molecular mechanisms involved in the ameliorative effect of L-lactate on LTP magnitude in AD mice require further in-depth studies.

It has been shown that learning memory activity increased the density and size of proximal dendritic spines of pyramidal neurons along with the PSD surface area, whereas these increases were abolished after inhibition of glycogen metabolism by injection of DAB before the learning memory activity and the coadministration of 100 nmol of *L*-lactate with 1000 pmol DAB, a combination that rescued the behavioral phenotype and increased spine density but failed to rescue the inhibition by DAB of spine volume and the PSD surface area increases evoked by learning [34]. It has also been shown that the bilateral hippocampal injection of lactate rescued the effect of DAB and restored a high profilin expression, which represents a relevant marker for spine morphological changes induced during long-term plasticity and memory [35]; this result indicated lactate is critical to enable the synaptic molecular and spine morphological changes underlying long-term memory formation. In our study, we found that the number of synapses in the hippocampus of early AD mice was significantly reduced, and the synaptic structure was impaired, as evidenced by the thickness of the PSD, and the width of the synaptic cleft was significantly reduced. Exogenous supplementation of *L*-lactate could rebuild the synaptic structure and ensure the structural basis of synaptic transmission.

Lactate is also critical for the induction of molecular changes required for memory formation. In our study, we found that supplementation with *L*-lactate increased the protein expression level of SYP. Interestingly, previous studies have shown that supplementation with lactate increased the protein expression of ARC [8,35], but our study found that 4 weeks after the injection of *L*-lactate, there was a significant decrease in ARC. ARC is an immediate early-expressed gene that can be induced rapidly. While previous studies examined 30 min after lactate injection, we were more interested in focusing on the expression of synaptic plasticity proteins after prolonged lactate supplementation rather than immediately, and thus found reduced expression of ARC in the hippocampus collected 48 h after the last *L*-lactate supplementation. In addition, ARC regulates the endocytosis of AMPA-type glutamate receptors (AMPARs), notch signaling, and spine size and type to affect the long-term potentiation and depression of synaptic transmission, homeostatic synaptic scaling, and adaptive functions, such as long-term memory formation [36]; however, in AD, it was involved in the generation of Aβ through interacting with presenilin 1 [37], so the paradoxical role of ARC in the regulation of synaptic plasticity or Aβ generation in AD, and whether *L*-lactate supplementation is involved in ameliorating the pathological process of AD through inhibition of ARC, require further in-depth studies.

## 4. Materials and Methods

### 4.1. Animals

In this study, 4-month-old APP/PS1/tau transgenic AD mice (AS, n = 80) and age-matched wild-type C57BL/6J mice (CS group, n = 35) were used. All mice were housed in a temperature- (22 ± 1C) and humidity-controlled room (50%) on a 12 h light/dark cycle (lights on 7:00) with ad libitum access to water and food and tested at 6 months of age. The animal experimental content and protocol were approved by the Experimental Ethics Committee for Sports Science of Beijing Sports University (approval number: 2019062A).

The 3xTg-AD mice received daily intraperitoneal injections of either vehicle (0.9% NaCl) (AN group, n = 25) or *L*-lactate solution (71718, Millipore-Sigma (St. Louis, MO, USA), 200 mg/kg body weight, dissolved in 0.9% saline for 30 days (AL group, n = 25)). Then, 48 h after the final injection, mice were used for behavioral tests or anesthetized with isofluran and sacrificed by decapitation; brain tissues were immediately frozen in liquid nitrogen, and these tissues were saved at −80 °C for subsequent analysis.

### 4.2. Eight-Arm Maze

An eight-arm maze (JLBehv-8RMG, JLsofttech, Shanghai, China) was used for assessing spatial memory ability. Each group of mice underwent 3 days of acclimatization to eliminate the stress response and explored freely for 10 min in all the arms where food was placed. At the end of acclimatization, only the target arm was placed with sweet food. Tests were performed once a day for 10 days during the test period, and each trial ended when the mouse had found all the food in the target arms. Measurements were made according to the frequency of working memory errors (the reentering of previously visited arms, indicated as short-term memory), the frequency of reference memory errors (entering an arm that was unbaited, long-term memory), and total exploration time. After each trial, the arms and central platform of the maze were cleaned to ensure that the mice could not follow their odor or that of other mice.

### 4.3. Step-Down Passive-Avoidance Test

The step-down passive avoidance test (ST-120, TECHMAN, Chengdu, China) consisted of an airtight testing case with a grid floor and a circular platform (4 cm diameter). During the training session, animals were placed on the platform; when the mice stepped down onto the floor, they would receive an electric shock (22V). Retention test sessions were carried out 48 h after training. Step-down latency on the test day was recorded as an index of inhibitory avoidance memory.

### 4.4. Lactate Concentration Detected

After the mice were anesthetized, the head was fixed, the skin of the neck was cut, and a drawn glass electrode was inserted into the endocranium of the mice. CSF was aspirated into the glass electrode, then collected into a centrifuge tube, and 10 μL was added to the EKF sample tube; the lactate content in CSF was measured using a lactate meter (Biosen C-line clinic, EKF, Stuttgart, Germany). After the execution of the mice, hippocampal tissue was taken, and the lactate content was determined according to the kit instructions (BC2235, Solarbio, Beijing, China).

### 4.5. Western Blot

Hippocampal tissues were homogenized in RIPA buffer containing protease and phosphatase inhibitors (Roche, Basel, Switzerland), and protein content was quantitated using a Pierce BCA Protein Assay kit (23227, ThermoFisher, Waltham, MA, USA). Then, equal amounts of proteins (24 μg) were separated by sodium dodecyl SDS–polyacrylamide gel electrophoresis (SDS-PAGE) and transferred to PVDF membranes, blocked for 2 h with 5% BSA, and then incubated overnight at 4 °C with antibodies against Arc (1:1000, Abcam, Cambridge, MA, USA), EGR1 (1:500, Abcam, USA), PSD95 (1:1000, Abcam, USA), and SYP (1:1000, Abcam, USA); membranes were washed and incubated with HRP-conjugated secondary antibody (1:5000, Proteintech, Wuhan, China). Signals were revealed using an ECL detection kit (1863095, ThermoFisher, USA), and the images were acquired using a chemiluminescence-measuring instrument (ChemiDoc XRS+, Bio-Rad, Hercules, CA, USA).

### 4.6. Brain Slice Preparation

Mice were anesthetized with isoflurane, followed by decapitation and hippocampus dissection, and embedded with 4% low melting point agarose. Hippocampal slices (400 μm) were cut in ice-cold modified sucrose solution (in mmol/L: 26 NaHCO_3_, 2.5 KCl, 1.25 NaH_2_PO_4_, 78 NaCl, 5 MgSO_4_·7H_2_O, 68 Sucrose, 25 Glucose) using a VT-1200S vibratome (Leica, Wetzlar, Germany). The slices were rinsed and transferred to a nylon mesh inside a storage chamber containing regular aCSF (in mmol/L: 3 KCl, 2 CaCl_2_·2H_2_O, 13 MgCl_2_·6H_2_O, 1.25 NaH_2_PO_4_, 119 NaCl, 25 NaHCO_3_, 10 Glucose) and were incubated for an additional 1 h at 25 ± 1 °C before recording. All solutions were saturated with 95% O_2_/5% CO_2_ (*v/v*).

### 4.7. LTP

For electrophysiological recordings, the slices were transferred into a recording chamber continuously perfused with aCSF (1.5 mL/min) at room temperature (28 ± 0.5 °C). CA3-CA1 areas were identified with differential interference contrast microscopy (IR-1000E, JASCO, Tokyo, Japan). Field excitatory postsynaptic potentials (fEPSPs) were recorded with a Multiclamp 700B amplifier (Axon, Manhattan, NY, USA) using pipettes (1–2 MΩ) filled with NaCl (2 mM) and placed in the stratum radiatum of the CA1 area, close to the subiculum. Synaptic potentials were evoked at 0.05 Hz by orthodromic stimulation (100 ms) of Schaffer collaterals using either a glass pipette filled with NaCl or a bipolar tungsten stimulating electrode placed in the stratum radiatum >200 mm away from the recording electrodes. LTP was induced by applying a high-frequency stimulation (HFS) protocol (100 Hz train of stimuli for 1 s, repeated four times at 20 s intervals) performed in the current clamp mode.

### 4.8. Transmission Electron Microscopy Analysis

Hippocampal tissues were fixed with 2.5% glutaraldehyde and washed three times in phosphate-buffered saline (PBS) (pH 7.2), following which they were post-fixed in 1% osmium tetroxide, dehydrated in graded ethanol and acetone solution, and embedded in epoxy resin. Sectioning and staining were subsequently performed under general electron microscopy supervision. The structure of synapses was observed and imaged under a transmission electron microscope (JEM1400, JEOL, Tokyo, Japan). The thickness of the postsynaptic density (PSD), the width of the synaptic cleft, and the synaptic curvature were determined as previously reported [38,39].

### 4.9. Statistics

All experimental data were expressed as mean ± standard deviation (Mean ± SD). Data were analyzed using SPSS software (25.0), and one-way ANOVA and Tukey’s post hoc test were used for comparison between the CS, AL, and AN after tests of normality and chi-squaredness. Independent samples *t*-tests were used to compare the experimental results of LTP in brain slices. Data were significantly different when *p* < 0.05.

## 5. Conclusions

In conclusion, this study identified a role for lactate in synaptic plasticity in early AD, by which *L*-lactate attenuated learning and memory deficits, synaptic structure, and function in 3xTg-AD mice. Thus, bolstering brain lactate levels may constitute a novel therapeutic strategy to protect and/or repair synapse impairment and ameliorate memory loss in early AD.

## Figures and Tables

**Figure 1 ijms-26-01486-f001:**
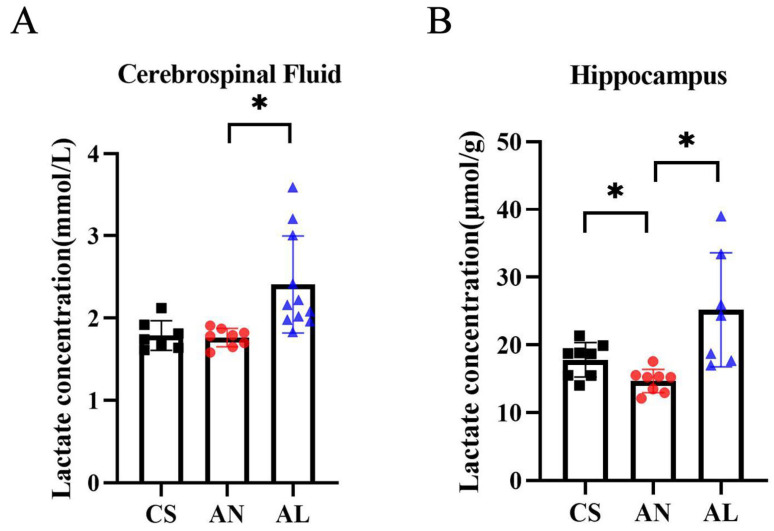
Exogenous *L*-lactate increased the concentration of brain lactate in AD mice. (**A**) Lactate concentration in CSF (CS: n = 7; AS: n = 8; AL: n = 11). (**B**) Hippocampal tissues (CS: n = 8; AN: n = 8; AL: n = 7). * *p*  <  0.05. Numerical data: CSF lactate (CS: 1.79 ± 0.18, AS:1.76 ± 0.11, AL: 2.41 ± 0.59), hippocampal lactate (CS: 17.82 ± 2.53, AN: 14.67 ± 1.74, AL: 25.17 ± 8.42).

**Figure 2 ijms-26-01486-f002:**
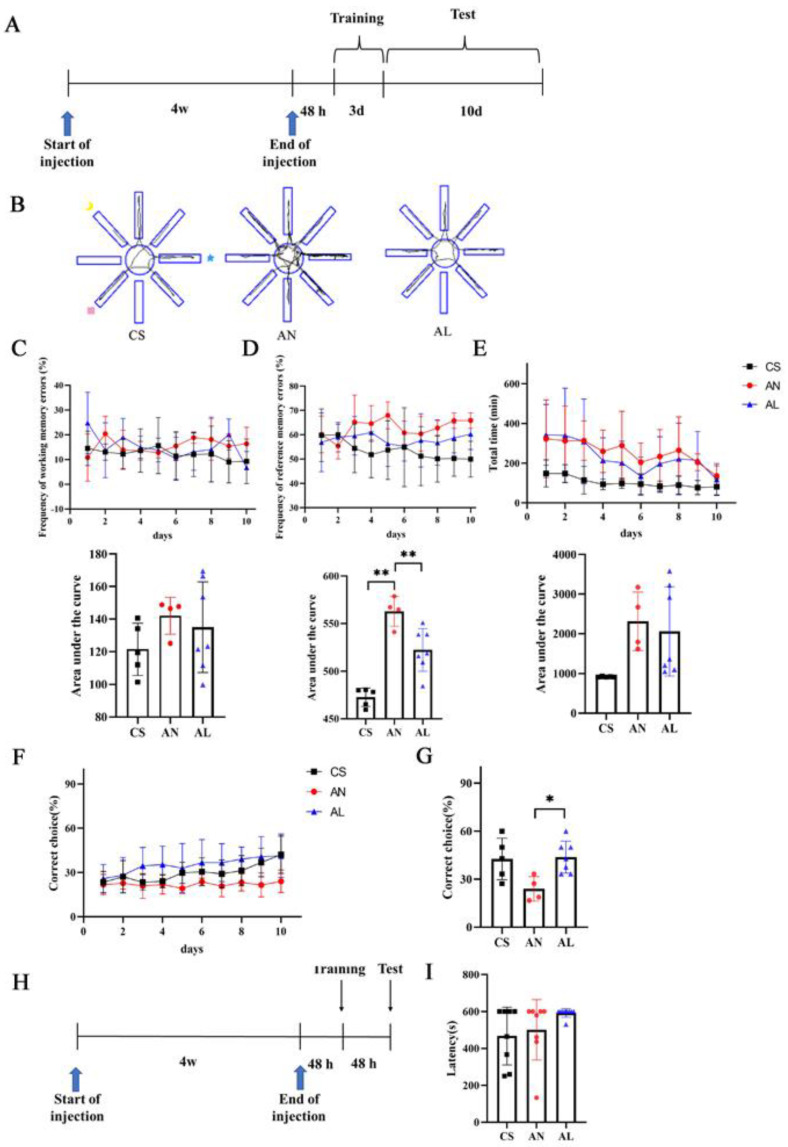
*L*-lactate improved the spatial memory of AD mice. (**A**) Experiment design of an eight-arm maze. (**B**) Representative tracks of mice moving in the eight-arm maze. (**C**–**E**) Trend of the frequency of working memory error, reference memory error, and exploration time over time and area under the curve (CS (n = 5), AN (n = 4), AL (n = 7)). (**F**) Trend of the percentage of the correct choice. (**G**) Bar graphs summarize the percentage of correct choices in the last trial. (**H**) Experiment design of a step-down passive-avoidance test. (**I**) Latency to find the platform (CS (n = 8), AN (n = 7), AL (n = 9)). * *p*  <  0.05, ** *p*  <  0.01. Numerical data: the area under the curve of the frequency of working memory errors (CS: 121.52 ± 15.97, AN: 142.03 ± 11.33, AL: 135.09 ± 27.77), the area under the curve of the frequency of reference memory errors (CS: 472.71 ± 9.64, AN: 562.93 ± 15.79, AL: 522.29± 22.34), the area under the curve of the exploration time (CS: 917.73 ± 10.63, AN: 2313.34 ± 736.71, AL: 2062.43 ± 1123.79), correct choice (CS: 42.69 ± 13.02, AN: 24.01 ± 7.72, AL: 43.86 ± 9.99), latency (CS: 467.50 ± 156.25, AN: 553.86 ±72.70, AL: 592.22 ± 24.75).

**Figure 3 ijms-26-01486-f003:**
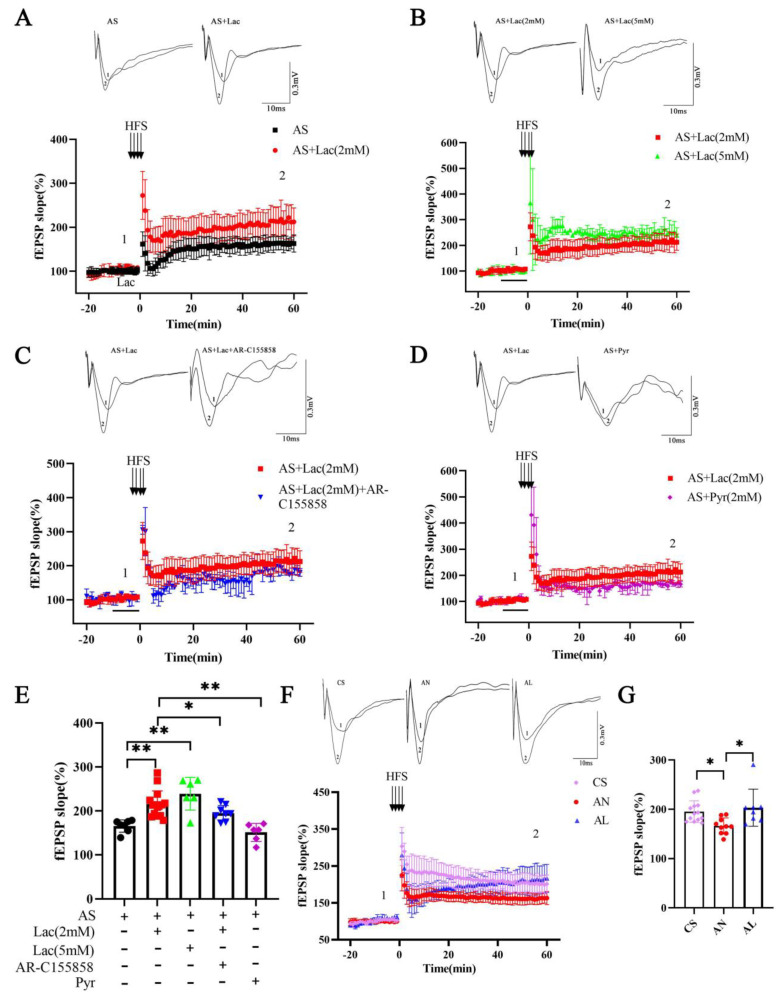
*L*-lactate-enhanced LTP magnitude in AD mice. (**A**–**D**) Maintenance of LTP after addition of *L*-lactate (2 mM, n = 6), *L*-lactate (5 mM, n = 3), *L*-lactate (2 mM) +AR-C155858 (1 μm) (n = 5), Pyruvate (2 mM, n = 3) to aCSF prior to high-frequency stimulation. (**E**) Bar graphs summarizing results presented in (**A**–**D**). (**F**) Maintenance of LTP in AD mice after intraperitoneal injection of *L*-lactate for 4 weeks (n = 5 mice per group). (**G**) Bar graphs summarizing results presented in (**F**). * *p * <  0.05, ** *p* <  0.01. Numerical data: fEPSP slope (AS: 165.50 ± 14.02, AS + Lac-2 mM: 215.41 ± 30.05, AS + Lac-5 mM: 238.95 ± 37.14, AS + Lac-2 mM + AR-C155858: 194.96 ± 17.33, AS + Pyr: 151.07 ± 20.53, CS: 195.23 ± 22.55, AN: 165.98 ± 16.32, AL: 203.21 ± 37.67).

**Figure 4 ijms-26-01486-f004:**
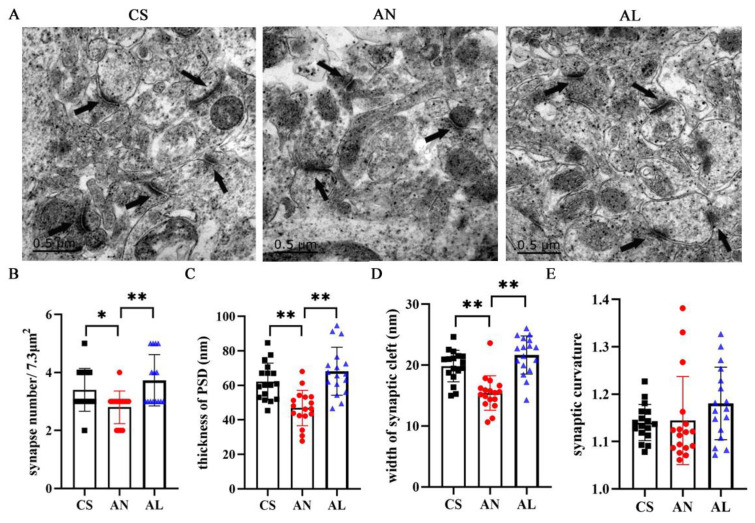
*L*-lactate improved the structure of synapses. (**A**) Transmission electron microscopy showed the ultrastructural features of synapses (black arrow) in the hippocampal CA1. (**B**–**E**) Bar graphs summarize the number of synapses and morphological measurement of the thickness of PSD, the width of the synaptic cleft, and synaptic curvature from synapses (n  =  three mice per group, five images per animal). * *p*  <  0.05, ** *p* <  0.01. Numerical data: synapse number (CS: 3.4 ± 0.74, AN: 2.8 ± 0.56, AL: 3.73 ± 0.88), thickness of PSD (CS: 62.15 ± 10.70, AN: 46.86 ± 10.23, AL: 68.18 ± 13.92), width of synaptic cleft (CS: 19.84 ± 2.59, AN: 15.42 ± 2.85, AL: 21.67 ± 3.13), synaptic curvature (CS: 1.14 ± 0.04, AN: 1.14 ± 0.09, AL: 1.18 ± 0.08).

**Figure 5 ijms-26-01486-f005:**
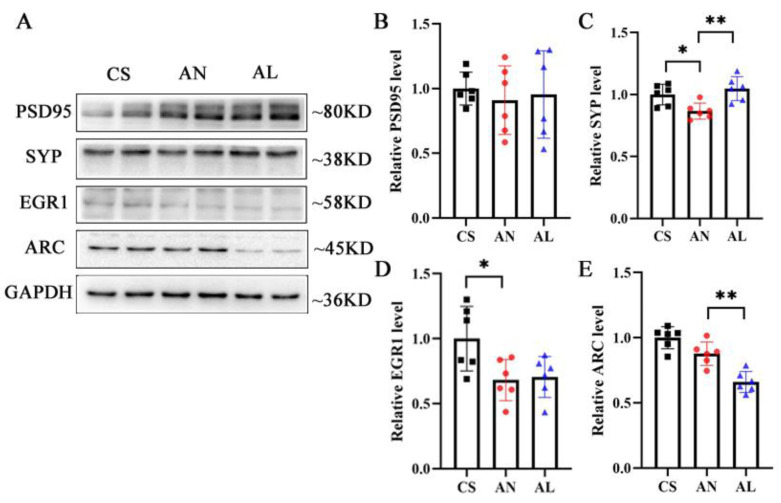
*L*-lactate increased the expression of synaptic plasticity proteins in AD mice. (**A**) Representative Western blot images of PSD95, SYP, EGR1, and ARC in CS, AN, and AL groups (n  =  six mice per group). (**B**–**E**) Quantitative depiction of PSD95, SYP, EGR1, and ARC protein levels in the hippocampus. * *p*  <  0.05, ** *p* <  0.01. Numerical data: relative PSD95 level (CS: 1.00 ± 0.13, AN: 0.91 ± 0.26, AL: 0.95 ± 0.34), relative SYP level (CS: 1.00 ± 0.08, AN: 0.87 ± 0.06, AL: 1.05 ± 0.10), relative EGR1 level (CS: 1.00 ± 0.25, AN: 0.68 ± 0.16, AL: 0.71 ± 0.16), relative ARC level (CS: 1.00 ± 0.08, AN: 0.88 ± 0.09, AL: 0.66 ± 0.08).

## Data Availability

The data presented in this study are available on request from the first author.

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
