# Peer review of "L*-Lactate Administration Improved Synaptic Plasticity and Cognition in Early 3xTg-AD Mice"

_ijms, 2025, doi:10.3390/ijms26041486_

Round 1

Reviewer 1 Report

Comments and Suggestions for Authors

In the manuscript titled “L-lactate administration improved synaptic plasticity and cognition in early 3xTg-AD mice”, the authors study the effects of l-lactate on different aspects of AD pathology including synaptic plasticity, morphological changes of the synapses and behavior. The authors provide interesting results indicating that interference with the energy supply, which is impaired in AD, improves synaptic plasticity, cognition-related tests, and molecular markers of synaptic structural homeostasis. The topic is of interest since metabolic arrestment is one of the pathophysiological hallmarks of AD. However, there are substantial improvements that authors should address before considering the manuscript adequate for publication in IJMS.

Major

1-      The manuscript is negligible for review but thinking in a final easily understandable version the manuscript should benefit from deep proofreading from a native English speaker or highly technical skilled colleague, particularly for grammar. (e.g. see lines 16, 26, 36-38, and so on…. The sentence in lines 62-65 is too long and can be divided into two sentences…).

2-      The authors should provide numerical data for each result, experiment or parameter (e.g., mean ± SEM or SD according to editorial guidelines and journal requirements), thus accounting for reproducibility and statistic checking. This data can be shown in the text when the result is mentioned or in the figure legend.

3-      In light of recent advances in studying AD pathology and how to counteract it in rodent models (e. g. see and refer to Arroyo-García et al., 2021 doi: 10.1038/s41380-021-01257-0,  Boza-Serrano et al., 2018 doi: 10.1038/s41598-018-19699-y), the authors may want to consider that 6 months is not too early since it is known that substantial pathological processes start very early far before preceding clinical onset (cognitive compromised state) and amyloid deposition or tau pathological aggregation.

4-      Please provide the complete terms for the acronyms (CA, AN, and AL) at the beginning of the results section the first time mentioned.

5-      Fig. 2G. There is a point that appears like an outlier. Authors should perform an outlier detection test to verify and clean out if needed, thus performing new statistics on these data that may lead to different conclusions.

6-      Line 87, figure caption. As it is written it looks like the passive avoidance test was also improved when data show that it was not impaired in the AD group.

7-      Line 93 needs to be properly supported by references. Please refer to and further build a discussion on Andrade-Talavera and Rodríguez-Moreno, 2021; doi: 10.3389/fnmol.2021.696476

8-      Line 95 and Figure 3. The data shown do not support that there is a deficit in LTP maintenance. Around 150% of magnitude is considered LTP indeed, at least early LTP. The authors should rephrase this sentence to conclude that LTP magnitude is enhanced in agreement with the results shown. It is also intriguing that this model displays LTP as it has been described before that LTP is impaired in this model. Please see, refer and build discussion upon the reference provided (Andrade-Talavera and Rodríguez-Moreno, 2021; doi: 10.3389/fnmol.2021.696476) where the authors deeply discuss differences between diverse studies on LTP in AD models including seminal studies in the 3Tg AD model.

9-      Figure 3F legend. Does each dot in the histogram bars represent one mouse? Please clarify. In this regard, in panel E it is expected that each symbol in the histogram bars corresponds to one brain slice. Is that correct? In addition, the effect of lactate and pyruvate over the baseline should be shown in an independent supplementary figure.  Importantly, in the figure legend, panel E states: 5 mice per group, but this is incongruent with the sample size shown in histogram bars in the corresponding symbols.

10-  Figure 4 legend. It is again a concern that according to the symbols placed on the histogram bars it appears to be 17 instead of 15 values for each measurement. The authors must carefully explain this mistake revising all the data for each figure and updating the real values of the sample sizes.

11-  Methods 4.7. Were the experiments performed at room temperature or at physiological temperature? It might affect the effect of metabolic interventions and the expression of synaptic plasticity. Clarify.

12-  Methods 4.9. It is highly recommended to show the statistics performed after each p-value either in the text or the author may choose to include it in the figure legends.

Minor

1.       Line 98. It is more conceptually correct to say magnitude instead of activity, please correct it throughout the text.

2.       Line 105. This sentence may benefit from proofreading for a better understanding of the take-home message.

3.       Line 209. Describe the acronym the first time used.

4.       Line 230. Maybe it is better to use healthy instead of normal.

Comments on the Quality of English Language

The manuscript is negligible for review but thinking in a final easily understandable version the manuscript should benefit from deep proofreading from a native English speaker or highly technical skilled colleague, particularly for grammar.

Author Response

Dear reviewer,

We sincerely thank you for your valuable feedback that we have used to improvethe quality of our manuscript. Your comments are laid out below in bold font and specific concernshave been numbered.

  1. The manuscript is negligible for review but thinking in a final easily understandable version the manuscript should benefit from deep proofreading from a native English speaker or highly technical skilled colleague, particularly for grammar. (e.g. see lines 16, 26, 36-38, and so on…. The sentence in lines 62-65 is too long and can be divided into two sentences…).

Answer: We sincerely appreciate the reviewer’s constructive feedback on improving the manuscript’s clarity and readability. We have now thoroughly revised the text to address grammatical inaccuracies and ensure proper sentence structure. Specifically: The manuscript has been professionally proofread by a native English-speaking colleague with expertise in neuroscience to refine grammar, syntax, and overall flow. The overly long sentence in lines 62–65 has been divided into two shorter, more digestible sentences to enhance readability (revised in lines 74–77 of the updated manuscript). Additional grammatical edits have been applied to flagged sections (e.g., lines 16, 30, 42-44) and throughout the text to eliminate ambiguities and align with standard academic conventions.We believe these revisions have significantly improved the clarity and precision of the manuscript while maintaining its scientific rigor.

  1. The authors should provide numerical data for each result, experiment or parameter (e.g., mean ± SEM or SD according to editorial guidelines and journal requirements), thus accounting for reproducibility and statistic checking. This data can be shown in the text when the result is mentioned or in the figure legend.

Answer: We thank the reviewer for emphasizing the importance of transparent data reporting to ensure reproducibility. In accordance with the journal’s editorial guidelines, we have now systematically revised the manuscript to include numerical data (mean ± SEM/SD) for all experimental results, parameters, and statistical analyses.

  1. In light of recent advances in studying AD pathology and how to counteract it in rodent models (e. g. see and refer to Arroyo-García et al., 2021 doi: 10.1038/s41380-021-01257-0,  Boza-Serrano et al., 2018 doi: 10.1038/s41598-018-19699-y), the authors may want to consider that 6 months is not too early since it is known that substantial pathological processes start very early far before preceding clinical onset (cognitive compromised state) and amyloid deposition or tau pathological aggregation.

Answer: We thank the reviewer for highlighting the importance of early pathological processes in Alzheimer’s disease (AD) and for directing us to the seminal studies by Arroyo-García et al. (2021) and Boza-Serrano et al. (2018). We fully acknowledge that AD-related pathological changes, including neuroinflammation and synaptic dysfunction, can emerge long before overt cognitive deficits or amyloid/tau pathology become detectable.

In our study, the selection of 6-month-old APP/PS1 mice was based on the following considerations:

Model-Specific Progression: While early pathological events (e.g., microglial activation, subtle synaptic alterations) may indeed occur prior to 6 months in certain AD models. The APP/PS1/Tau triple-transgenic mouse is a classic Alzheimer's disease (AD) model that exhibits age-dependent pathological features characteristic of AD. At 6 months of age, these mice develop amyloid-beta (Aβ) plaque deposits in the posterior hippocampus, accompanied by hyperphosphorylated tau protein and potential cognitive impairment—which aligns with our focus on investigating interventions during the prodromal stage of pathology. (e.g., Ramona Belfiore et al., 2020; DOI: https://doi.org/10.1111/acel.12873)

Clinical Relevance: This time point models a critical window where therapeutic strategies might still rescue or delay functional decline, as opposed to intervening after irreversible neurodegeneration.

Consistency with Prior Literature: Our choice is supported by previous studies utilizing similar age ranges in APP/PS1/tau mice to evaluate synaptic and cognitive rescue mechanisms (e.g., Le Douce et al., 2020; DOI: https://doi.org/10.1016/j.cmet.2020.02.004).

Previous studies from our laboratory demonstrated that 2-month-old 3×Tg AD mice exhibited no significant pathological alterations in long-term potentiation (LTP). However, as the disease progressed to 6 months of age, synaptic transmission efficacy and LTP in the hippocampal CA1 region were markedly impaired.

  1. Please provide the complete terms for the acronyms (CA, AN, and AL) at the beginning of the results section the first time mentioned.

Answer: We sincerely appreciate the reviewer’s attention to clarity in terminology. In accordance with the journal’s style guidelines, we have revised the manuscript to explicitly define all acronyms upon their first mention in the Results section. 

  1. 2G. There is a point that appears like an outlier. Authors should perform an outlier detection test to verify and clean out if needed, thus performing new statistics on these data that may lead to different conclusions.

Answer: We thank the reviewer for their meticulous attention to data integrity. Upon performing outlier detection, we identified one outlier in the AN group. After removing this outlier, we re-ran the statistical analysis. The revised results demonstrated that no significant differences between groups. 

  1. Line 87, figure caption. As it is written it looks like the passive avoidance test was also improved when data show that it was not impaired in the AD group.

Answer: We sincerely appreciate the reviewer’s careful reading. We have revised the figure caption  to“L-lactate improved spatial memory of AD mice”

  1. Line 93 needs to be properly supported by references. Please refer to and further build a discussion on Andrade-Talavera and Rodríguez-Moreno, 2021; doi: 10.3389/fnmol.2021.696476

Answer: Thank you for your valuable suggestions. We have integrated this reference into the main text and supplemented the following additional content: Early studies have demonstrated that HFS protocols are used to induce LTP to mimic the synchronized firing of neuronal ensembles observed during natural learning processes and are directly involved in memory encoding. As such, LTP has been widely adopted as a canonical cellular model of memory formation, serving as a quintessential experimental paradigm for investigating synaptic plasticity[18].

  1. Line 95 and Figure 3. The data shown do not support that there is a deficit in LTP maintenance. Around 150% of magnitude is considered LTP indeed, at least early LTP. The authors should rephrase this sentence to conclude that LTP magnitude is enhanced in agreement with the results shown. It is also intriguing that this model displays LTP as it has been described before that LTP is impaired in this model. Please see, refer and build discussion upon the reference provided (Andrade-Talavera and Rodríguez-Moreno, 2021; doi: 10.3389/fnmol.2021.696476) where the authors deeply discuss differences between diverse studies on LTP in AD models including seminal studies in the 3Tg AD model.

Answer: We sincerely appreciate the reviewer’s insightful critique and constructive suggestions.  We have carefully revised the manuscript to address these concerns as follows:

In line with the reviewer’s observation, we agree that the original phrasing regarding "LTP maintenance deficits" could be misinterpreted. We have rephrased this section to state: "L-lactate treatment significantly enhanced LTP magnitude". This revision aligns with the reviewer’s recommendation and emphasizes the observed enhancement rather than a deficit.

  1. Figure 3F legend. Does each dot in the histogram bars represent one mouse? Please clarify. In this regard, in panel E it is expected that each symbol in the histogram bars corresponds to one brain slice. Is that correct? In addition, the effect of lactate and pyruvate over the baseline should be shown in an independent supplementary figure.  Importantly, in the figure legend, panel E states: 5 mice per group, but this is incongruent with the sample size shown in histogram bars in the corresponding symbols.

Answer: Figure 3E: Each symbol in the histogram bars corresponds to one brain slice. Figure 3F: Average traces of field excitatory postsynaptic potentials (fEPSPs) in hippocampal slices from the three experimental groups, where each point represents the mean fEPSP value per minute for a brain slice.

The purpose of recording the LTP activity following lactate and pyruvate administration was to demonstrate that pyruvate, although serving as an energy substrate, fails to exert the same restorative effects as lactate. This contrast further highlights the unique role of lactate in improving synaptic plasticity.

Additionally, in the LTP experiments shown in Figure 3E, each group utilized 5 mice, with 1-3 brain slices recorded per mouse. The data points in Figure 3F correspond to the total number of brain slices analyzed.

  1. Figure 4 legend. It is again a concern that according to the symbols placed on the histogram bars it appears to be 17 instead of 15 values for each measurement. The authors must carefully explain this mistake revising all the data for each figure and updating the real values of the sample sizes.

Answer: In the electron microscopy experiments, each group included 3 mice, and 5 images were captured per mouse, resulting in 15 data points per group in Figure 4B. These points represent the synaptic counts analyzed across the 15 images. However, in Figures 4C, 4D, and 4E, each data point corresponds to individual synapses. As noted, a single image may contain multiple synapses, and we analyzed more than one synapse per image when possible. This approach was adopted to increase the sample size and ensure a more robust representation of the results. Consequently, the number of data points in the histograms exceeds 15, reflecting the total number of synapses quantified rather than the number of images.

  1. Methods 4.7. Were the experiments performed at room temperature or at physiological temperature? It might affect the effect of metabolic interventions and the expression of synaptic plasticity. Clarify.

Answer: We sincerely appreciate the reviewer's insightful question. We had indeed considered the importance of physiological temperature in mimicking in vivo conditions, where synaptic activity and metabolic states are more authentically preserved. However, due to experimental constraints, particularly the need to prolong slice viability during extended LTP recordings (typically 60–90 minutes), we conducted the experiments at room temperature (28±0.5°C).

While we acknowledge that lower temperatures may modestly influence metabolic rates and synaptic transmission kinetics, our pilot studies confirmed that LTP induction and maintenance remained robust under these conditions. We have revised Methods section (Section 4.7). In future electrophysiological experiments, we plan to implement advanced temperature-controlled systems to further validate our findings under physiological conditions (34–37°C).

  1. Methods 4.9. It is highly recommended to show the statistics performed after each p-value either in the text or the author may choose to include it in the figure legends.

Answer: We sincerely appreciate the reviewer's valuable suggestion regarding the presentation of statistical details. We have shown the p-value in the text.

Minor

  1. Line 98. It is more conceptually correct to say magnitude instead of activity, please correct it throughout the text.

Answer: We appreciate this critical conceptual clarification. We have systematically replaced "activity" with "magnitude" when describing LTP measurements to better reflect the quantitative nature of synaptic potentiation. 

  1. Line 105. This sentence may benefit from proofreading for a better understanding of the take-home message.

Answer: We have thoroughly revised the indicated sentence to enhance clarity and scientific rigor. The updated text now explicitly states: “Mirroring these in vitro findings, chronic in vivo L-lactate supplementation (4 weeks) significantly improved LTP magnitude in 6-month-old 3xTg-AD mice compared to AD controls(P=0.013) (Figures 3F, G).”

  1. Line 209. Describe the acronym the first time used.

Answer: We sincerely apologize for this oversight. The full term "1,4-dideoxy-1,4-imino-D-arabinitol(DAB)" has been added upon its first mention. Additionally, we have conducted a full-text review to ensure all acronyms are properly defined at their first occurrence.

  1. Line 230. Maybe it is better to use healthy instead of normal.

Answer: We fully agree that "healthy" is a more precise and inclusive descriptor. The term "normal mice" has been replaced with "healthy mice" throughout the manuscript. 

Reviewer 2 Report

Comments and Suggestions for Authors

The study is well-designed, and the authors effectively demonstrate lactate's protective role in reversing cognitive and synaptic deficits associated with Alzheimer's disease in APP/PS1 mice. I recommend that the authors revise the introduction to emphasize the novel aspects of the study. Additionally, since lactate exhibits an inverted U-shaped effect, it is important for the authors to describe and compare the dosage of lactate used in this study with those reported in previous research (see https://doi.org/10.1016/j.phrs.2024.107357; https://doi.org/10.7554/eLife.85751).

Furthermore, the methods and data presented for the radial arm maze require enhancement. The authors should specify how many trials were conducted in a day and the interval between trials. Utilizing the area under the curve does not convey sufficient information; therefore, I suggest calculating the percentage of correct choices instead of AUC and presenting this graphically. Please refer to the following reference for the radial arm maze protocol (PMID: 28495605). The electrophysiology experiments were conducted exceptionally well.

I believe the manuscript could be accepted once the above points are addressed.

Comments on the Quality of English Language

Could be improved

Author Response

Dear reviewer,

We sincerely thank you for your valuable feedback that we have used to improvethe quality of our manuscript. Your comments are laid out below in bold font and specific concernshave been numbered.

The study is well-designed, and the authors effectively demonstrate lactate's protective role in reversing cognitive and synaptic deficits associated with Alzheimer's disease in APP/PS1 mice. I recommend that the authors revise the introduction to emphasize the novel aspects of the study. Additionally, since lactate exhibits an inverted U-shaped effect, it is important for the authors to describe and compare the dosage of lactate used in this study with those reported in previous research (see https://doi.org/10.1016/j.phrs.2024.107357; https://doi.org/10.7554/eLife.85751).

Furthermore, the methods and data presented for the radial arm maze require enhancement. The authors should specify how many trials were conducted in a day and the interval between trials. Utilizing the area under the curve does not convey sufficient information; therefore, I suggest calculating the percentage of correct choices instead of AUC and presenting this graphically. Please refer to the following reference for the radial arm maze protocol (PMID: 28495605). The electrophysiology experiments were conducted exceptionally well.

I believe the manuscript could be accepted once the above points are addressed.

Answer: We extend our deepest gratitude for your expert guidance, which has substantially enhanced the scientific value and methodological rigor of our work. Your recommended literature has been systematically integrated into the revised manuscript. All constructive suggestions have been meticulously addressed through comprehensive text revisions as detailed in our point-by-point response letter. We have addressed each point as follows:

  1. Introduction Revision (Last paragraph)

Accumulating evidence suggests that lactate supplementation may ameliorate cognitive deficits induced by brain injury[14] and exert antidepressant effects[15]. However, in the context of AD, acute exogenous lactate administration has demonstrated limited efficacy in improving global cognitive function among AD patients, with only marginal enhancements observed in semantic memory[16,17]. Given that the early stages of AD represent a critical therapeutic window for intervention, it remains unclear whether chronic lactate supplementation during this pivotal phase could mitigate cognitive impairments associated with aerobic glycolysis dysfunction. Thus, in this study, we investigated whether lactate could improve cognition, synaptic structure and function in early 3xTg-AD mice and tested the hypothesis that lactate may be a potential target for the prevention and treatment of AD by improving synaptic plasticity and memory.

  1. Lactate Dosage Comparison and Inverted U-Shaped Effect

In this this study, we explored whether the L-lactate concentration effect could influence LTP magnitude, the results showed that 5 mM L-lactate did not have a significant difference than 2mM. However, it has been shown that elevated lactate(10mM) reversibly suppressed firing of hippocampal pyramidal cells[29], and resulted in suppression of gamma oscillations and attenuation of synaptic transmission, reduction of neurotransmitter release[30].

We think that inverted U-shaped trajectory of lactate dynamics in AD mechanistically linked to stage-specific pathological alterations. During early AD pathogenesis, impaired cerebral energy metabolism—particularly the disruption of astrocytic aerobic glycolysis—leads to diminished lactate production[27], thereby exacerbating neuronal bioenergetic deficits. As AD progresses, hyperactivated microglia drive a pathological cascade, inducing reactive astrocyte transformation with amplified glycolytic flux (astrocytic metabolic reprogramming), which culminates in aberrant lactate overproduction[28]. This biphasic regulatory pattern may reflect stage-specific interactions between glial metabolic reprogramming and neurodegenerative progression, offering mechanistic insights into the dual roles of lactate in AD pathophysiology. This biphasic regulatory mechanism, now incorporated into our discussion,

  1. Radial Arm Maze Method and Data Revision

Eight-arm maze (JLBehv-8RMG, JLsofttech, China) was used for assessing spatial memory ability. Each group of mice underwent 3 days of acclimatization to eliminate the stress response and explored freely for 10 min in all the arms where food was placed. At the end of acclimatization, only the target arm was placed with sweet food. Tests were performed once a day for 10 days during the test period, and each trial ended when the mouse had found all the food in the target arms. Measurements were made using the frequency of working memory errors (reentering of previously visited arms, indicated as short-term memory), frequency of reference memory errors (entering an arm that was unbaited, long-term memory) and total exploration time. After each trial, the arms and central platform of the maze were cleaned to ensure that the mice could not follow their own odor or that of other mice.

Round 2

Reviewer 1 Report

Comments and Suggestions for Authors

The authors have addressed all my concerns

Reviewer 2 Report

Comments and Suggestions for Authors

The authors have revised the manuscript asp er suggestions.